# Correction of a Factor VIII genomic inversion with designer-recombinases

Felix Lansing [1], Liliya Mukhametzyanova[1], Teresa Rojo-Romanos [1], Kentaro Iwasawa[2,3], Masaki Kimura[2], Maciej Paszkowski-Rogacz [1], Janet Karpinski[1], Tobias Grass[1,4], Jan Sonntag[1], Paul Martin Schneider [1], Ceren Günes [5], Jenna Hoersten [1], Lukas Theo Schmitt [1], Natalia Rodriguez-Muela[4], Ralf Knöfler[6], Takanori Takebe [2,3,7] & Frank Buchholz [1✉]

Despite advances in nuclease-based genome editing technologies, correcting human disease-causing genomic inversions remains a challenge. Here, we describe the potential use of a recombinase-based system to correct the 140 kb inversion of the F8 gene frequently found in patients diagnosed with severe Hemophilia A. Employing substrate-linked directed molecular evolution, we develop a coupled heterodimeric recombinase system (RecF8) achieving 30% inversion of the target sequence in human tissue culture cells. Transient RecF8 treatment of endothelial cells, differentiated from patient-derived induced pluripotent stem cells (iPSCs) of a hemophilic donor, results in 12% correction of the inversion and restores Factor VIII mRNA expression. In this work, we present designer-recombinases as an efficient and specific means towards treatment of monogenic diseases caused by large gene inversions.

[1] Medical Systems Biology, Medical Faculty, Technical University Dresden, 01307 Dresden, Germany. [2] Division of Gastroenterology, Hepatology and Nutrition, Division of Developmental Biology, Center for Stem Cell and Organoid Medicine (CuSTOM) Cincinnati Children's Hospital Medical Center, Cincinnati, OH, USA. [3] Department of Pediatrics, University of Cincinnati College of Medicine, Cincinnati, OH, USA. [4] German Center for Neurodegenerative Diseases, Helmholtz Association, 01307 Dresden, Germany. [5] Department of Cell and Developmental Biology, Max Planck Institute for Molecular Biomedicine, Röntgenstrasse 20, Münster 48149, Germany. [6] Department of Pediatric Hematology and Oncology, University Hospital Dresden, 01307 Dresden, Germany. [7] Institute of Research, Tokyo Medical and Dental University (TMDU), Tokyo, Japan. ✉email: frank.buchholz@tu-dresden.de

Today, patients with monogenic diseases are primarily treated with palliative care. However, within the past decades, disease-causing mutations have been identified in over 8000 monogenic diseases allowing for the development of cure strategies that target the underlying genetic defects[1,2]. Nuclease-based genome editing approaches can be used to correct the mutations for many of these genetic diseases[3–5]. Nevertheless, editing of large genomic alterations causative for genetic disorders, such as gene inversions, remains a challenge. Progress has been reported to correct gene inversions with nuclease-based approaches[6,7] but there are several drawbacks hampering the applied use of nucleases for these genetic alterations.

In order to repair DNA inversions with nucleases, two cuts have to be introduced into the genome, releasing the DNA fragment and subsequently relying on the fragment reinserting in the opposite orientation. This reversion is caused passively by the cell's DNA repair machinery and not actively by the nuclease itself. Therefore, editing efficiencies rely on an effective DNA repair machinery, which is most active in mitotic cells and typically less efficient in postmitotic cells[8]. Most certainly a nuclease-based treatment to correct inversions will require a cell-based therapy, where the correction and screening of the patient-derived cells would have to be performed ex vivo[7]. Additionally, it is largely unpredictable how the cell is going to repair the introduced lesions, providing a risk for sequence alterations such as small insertions/deletions (indels), thereby increasing the likelihood of unwanted mutations after the genome editing process. Furthermore, nuclease-based edited clones may also carry other undesirable small and large genomic rearrangements[5,9,10].

An alternative genome engineering tool with the potential to invert DNA in mitotic and postmitotic cells are site-specific recombinases such as the tyrosine recombinase Cre/loxP system[11,12]. Cre can precisely invert genomic sequences in model organisms when the loxP target sites are aligned on the same chromosome in an inverted orientation[13]. Since the inversion reaction performed by Cre is independent of cellular processes (e.g., DNA repair), it represents a promising alternative to nucleases[11]. To overcome the native target preference of Cre, methods of directed molecular evolution can be applied to develop novel Cre-type recombinases to recognize and actively recombine desired target sites[14,15], with first examples of evolved Cre-type recombinases demonstrating their broader potential for therapeutic applications[16,17].

We therefore reasoned that designer-recombinases could be suitable to correct genomic inversions involved in monogenic diseases. To test this hypothesis, we decided to evolve recombinases targeting sequences that are implicated in a disease-causing inversion of the F8 gene (ID: 2157, Homo sapiens). The resulting disease is a severe form of a blood clotting defect (Hemophilia A) where no Factor VIII activity is detectable. This inversion is caused by an intrachromosomal recombination event of two homologous regions found on the X-chromosome. One repeat is located in intron 1 (int1h-1) of the F8 gene and the other is located 140 kb upstream (int1h-2), outside the F8 gene. This inversion displaces exon 1 roughly 140 kb upstream of the F8 gene and fully disrupts Factor VIII expression[18,19]. Restoring 2–5% of the Factor VIII activity is thought to be sufficient to alleviate severe Hemophilia A symptoms[20], indicating that methods able to correct a moderate number of cells in the body could potentially result in clinical benefits.

Here we show the independent evolution of two designer-recombinases that cooperatively recombine the loxF8 target site found in the inverted repeats around the exon 1 of the F8 gene. Physical linking of the individual monomers resulted in a more specific recombinase system (RecF8) without compromising its on-target activity. A one-time delivery of RecF8 as mRNA to endothelial cells, differentiated from hemophilic patient-specific induced pluripotent stem cells, corrected the 140 kb inversion of the F8 gene in ~12% of the cells. This genomic correction led to ~6% F8 mRNA expression.

## Results

**Identification of loxF8 target site and directed evolution of F8 recombinases.** The int1h homologous regions are nearly perfect inverted repeats of 1041 bp harboring only one mismatch. Therefore, almost any target site for site-specific recombinases within these homologous regions would theoretically qualify as an inversion substrate for the recombination reaction. Typical target sites for Cre-type recombinases consist of 13 bp inverted repeats flanking an 8 bp spacer sequence[11]. The ideal target sequence for Cre-type recombinases would consist of a perfect inverted repeat left and right from the central spacer. This perfect sequence pattern does not exist within the int1h-repeats. However, evolved Cre-type recombinases can be coaxed to recombine target sites with some asymmetry[16,17]. Therefore, the int1h sequences were searched for target sites with tolerable asymmetry between possible half-sites. Systematically searching through the 1041 bp int1h sequences revealed 82 potential target sites (Fig. 1a, Supplementary Fig. 1, Supplementary Data 1). We chose the candidate sequence with the highest score (hereafter referred to as loxF8) of this list as a target for the evolution of Cre-type recombinases. The loxF8 site has 6 asymmetric positions and is located between base pairs 76-109 in the int1h regions (Fig. 1a).

New recombinases to target loxF8 were generated using the well-established substrate-linked directed evolution method (SLiDE, Fig. 1b)[14,16,17]. SLiDE works by linking recombinase activity to the plasmid DNA encoding the enzyme. Active recombinases can be easily selected by restriction enzyme digestion followed by a PCR reaction that specifically amplifies desirable recombinase variants (Fig. 1b). SLiDE was guided through different subsites to progressively evolve new recombinase variants toward the final target site. Progress can be monitored by plasmid-based restriction enzyme digestion after each evolution cycle. (Fig. 1b, Supplementary Figs. 2 and 3).

Selecting an asymmetric target site provides the opportunity to compare two different evolution strategies. A single recombinase can be evolved to recognize both 13 bp half-sites or two recombinases can be evolved in parallel for each 13 bp half-site (Fig. 1c, Supplementary Fig. 2)[17,21]. Combining two recombinases allows forming a functional heterodimer capable of recombining an asymmetric site[21]. To compare the two approaches, we performed two parallel SLiDE experiments in E. coli to either generate single or dual-recombinases that recombine loxF8 (Fig. 1d, Supplementary Figs. 2 and 3). For the single recombinase, 6 subsites and 140 rounds of SLiDE were required to obtain recombinase libraries with activity on the loxF8 target site (Supplementary Fig. 2).

In order to evolve the dual-recombinases to cooperatively recombine the loxF8 site as heterodimers, two separate evolutions were initiated. Two recombinase libraries were evolved for either the left (loxF8-L) or the right (loxF8-R) half-site of the loxF8 site (Supplementary Fig. 3). SLiDE evolution through 3 subsites each (Supplementary Fig. 3) and 115 rounds of directed evolution generated two libraries with activity on loxF8-L or loxF8-R, respectively. Hence, 25 evolution cycles less were required for the dual-recombinase approach. Importantly, only upon co-expression of both recombinase libraries, efficient recombination occurred on the asymmetric loxF8 target site, demonstrating that the participation of both monomers was required to recombine loxF8. Moreover, the libraries specific for loxF8-L or loxF8-R did not show any cross-reactivity, nor did they recombine the final

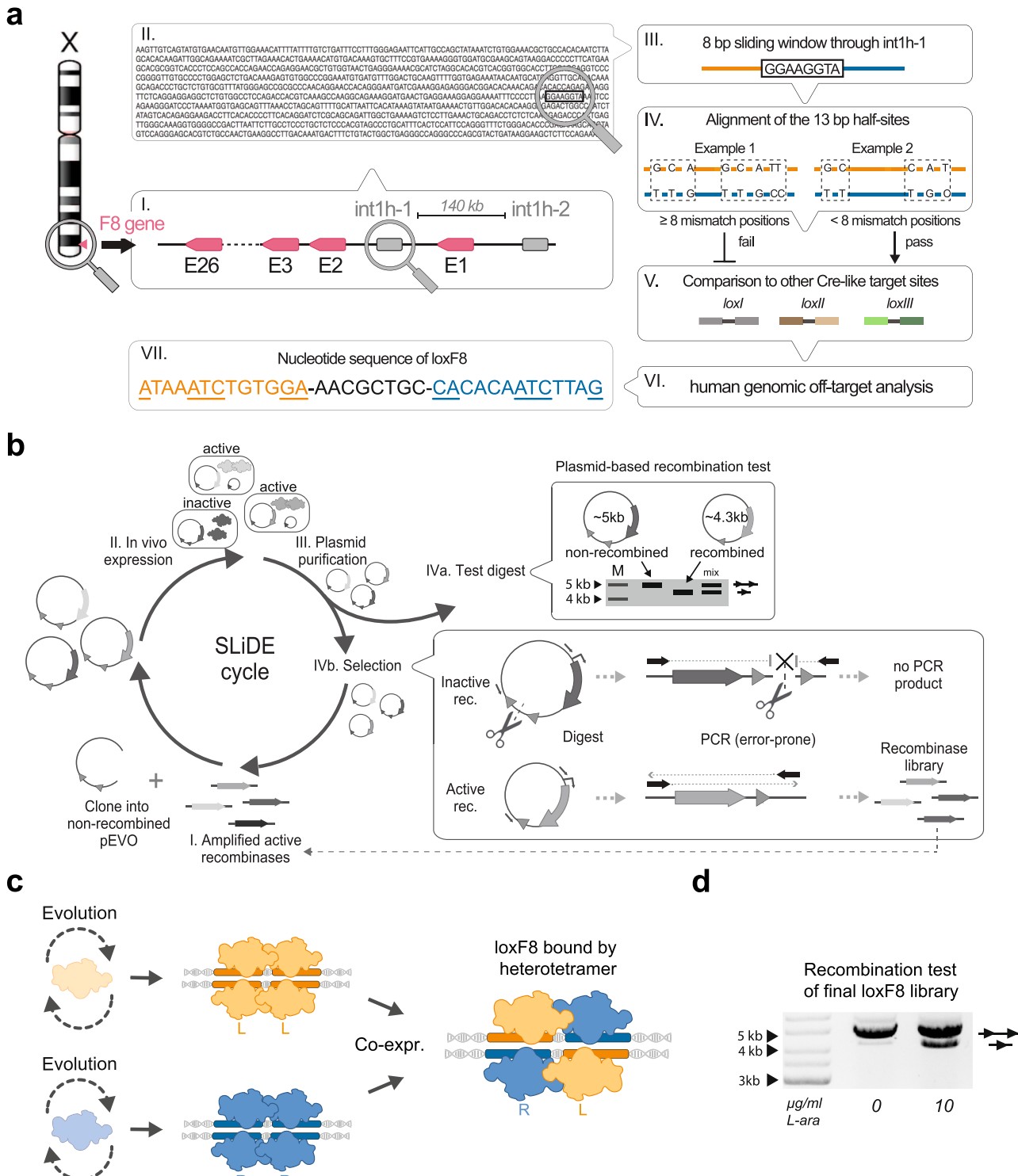

loxF8 target site when expressed as monomers (Supplementary Fig. 4). Because the recombination activity was higher for the recombinase heterodimers than for the recombinase monomers, we continued with the heterodimer libraries for further experiments.

**Selection of a recombinase heterodimer**. In order to select a candidate recombinase heterodimer, we screened for recombinases that were well tolerated in human cells. The loxF8-L and loxF8-R libraries were expressed for 21 days in HeLa cells utilizing a doxycycline inducible lentiviral system. Recombinases

were subsequently retrieved from genomic DNA and their activity was confirmed in *E. coli* (Fig. 2a). Importantly, the retrieved recombinase libraries were again only active on the final loxF8 target site when expressed as a dimer and showed no cross-reactivity when expressed as monomers (Supplementary Fig. 5). Next, 96 single heterodimer clones were analyzed by PCR for their activity on the loxF8 site (Supplementary Fig. 6). Four clones (D4, D7, D9 and D10) were analyzed in a plasmid-based recombination assay in more detail for their dose dependent activity (Supplementary Fig. 6). The best clone, D7, showed induction dependent recombination activity on the loxF8 target

**Fig. 1 Directed evolution of site-specific recombinases for the loxF8 target site. a** Overview of the target site identification in the inverted repeats (int1h) surrounding the F8 gene. For each 8 bp sequence in int1h-1 the surrounding 13 bp were compared. Every 13 bp sequence pair with less than 8 mismatch positions was analyzed for similarity to previous target sites targeted by evolved Cre-type recombinases. After off-target analysis the loxF8 target site was nominated. Underlined nucleotides display asymmetric positions. The nucleotide sequence of the left (orange) and the right (blue) half-sites are indicated. **b** Overview of substrate-linked directed evolution (SLiDE). The evolution starts by cloning recombinase libraries into the pEVO expression vector (I.). Single recombinase genes are displayed in different gray scales. Two lox-like sites for the recombinases are indicated as small triangles (excision orientation). After expression (II.) of the recombinases, plasmids are analyzed (III.). A library activity assay is performed for each cycle (IVa.). A schematic representation of the different recombination products on an agarose gel is shown. Marker and sizes are indicated. Active recombination variants that are further evolved (IVb.). Upon recombination a unique restriction site (scissors) is excised. Applying a restriction digest will linearize the non-recombined plasmid, recombined plasmid remains circular. A PCR (error-prone) using the indicated primers (arrows) will only generate a product from recombined plasmids. This amplified and mutated active recombinase variants are then subjected to the next evolution round. **c** Strategy to evolve a heterodimer of recombinases to target the loxF8 site. The monomers of the heterodimer recognize either the left (loxF8-L, orange) or the right (loxF8-R, blue) half-site of the loxF8 target site. Recombinases are shown as homo- or heterotetramers binding to two target sites in the recombination synapses. **d** Plasmid-based activity assay of the recombinase heterodimer library evolved for targeting loxF8. The agarose gel picture of the final heterodimer library recombining the loxF8 site is shown at two different expression levels (0 and 10 μg/ml L-arabinose). The upper band represents the unrecombined plasmid (illustrated by a line with two triangles), with the lower band showing the recombined plasmid (a line with one triangle). Marker (M) and sizes are indicated. Source data are provided as a Source data file.

site with nearly full recombination observed at 100 μg/ml L-arabinose, comparable with activity profiles of Cre on loxP or Brec1 on loxBTR[17] (Fig. 2b). Based on this promising profile, D7 was selected as a candidate for detailed studies in human cells.

The overall comparison of the amino acid sequence of the two D7 monomers to Cre revealed 46 changes (13%) in the left monomer and 39 changes (11%) in the right monomer (Fig. 2c), suggesting that considerable changes in the protein sequence were necessary to achieve high activity on the new target site. Interestingly, some of the common changes have previously been found in evolved Cre-type recombinases (Y77H, S108G, I166V, A175S, N317T, and I320S—Supplementary Fig. 6)[16,17,21], indicating that these residues might be beneficial for evolved Cre-type recombinases in general, or that these changes are potentially important for the stability of the enzymes, as suggested by a previous study[22].

Plotting the amino acid differences found in D7 onto a co-crystal structure of Cre bound to loxP[23], showed that the majority of the altered amino acids reside in proximity to DNA (Supplementary Fig. 7), consistent with their role in influencing DNA-binding specificity[11,24]. Both monomers show extensive amino acid changes in the J-helix, which interacts with the DNA major grove and is important for binding of Cre to loxP[24] (Fig. 2c and Supplementary Fig. 7). Interestingly, amino acid residues (Arg259, Glu262 and Glu266) that interact with the DNA target site region around nucleotide position 26–30 were mutated to different residues in the monomers (Supplementary Fig. 7). These changes might have evolved because the loxF8-L and loxF8-R target sites have different nucleotides in this region compared to loxP. Moreover, other amino acids known to interact with DNA[11] specifically changed in the individual monomers (Supplementary Fig. 7). Altogether, the mutational data provide a starting point for a more rational understanding for DNA-binding specificity of designer-recombinases for future studies.

**D7 recombines loxF8 in human cells**. In order to test the activity of the D7 heterodimer in human cells, a HEK293T^loxF8 reporter cell line was generated (Supplementary Fig. 8), which expresses the fluorescent mCherry protein after successful recombination of the loxF8 target sites (Fig. 3a). HEK293T^loxF8 cells were co-transfected with mRNA coding for the recombinases together with tagBFP mRNA (Supplementary Fig. 9), to evaluate whether mRNA delivery is a suitable method to express recombinases in human cells. Transfection of HEK293T^loxF8 with either D7-L or D7-R alone did not result in relevant numbers of mCherry positive cells, whereas many mCherry positive cells were observed

when both monomers were co-transfected, demonstrating that heterodimer formation is required to achieve efficient recombination (Fig. 3b, c). Analysis of the flow cytometry data showed that the heterodimer recombined ~76% of transfected HEK293-T^loxF8 reporter cells, in line with results obtained with Cre transfected into HEK293T^loxP reporter cells (Fig. 3c, Supplementary Fig. 10), signifying the robust recombination efficiency achieved by mRNA-mediated recombinase delivery. Importantly, PCR assays followed by DNA sequencing confirmed precise recombination reactions (Supplementary Fig. 10).

Because HEK293T^loxF8 cells carry the F8 gene in the wild type orientation, recombinase expression could potentially invert the loxF8 locus into the disease orientation as it is present in patients with severe Hemophilia A caused by the F8 int1h inversion. To test for this possibility, genomic DNA was extracted from the transfected cells and analyzed for the conceivable genomic inversion of the loxF8 locus. Two PCR assays were designed at the inversion sites to detect the orientation of the DNA fragment between the endogenous loxF8 target sites (Fig. 3d). Genomic DNA from a male donor with the F8 int1h inversion was used as a positive control. Indeed, only the HEK293T^loxF8 cells expressing the D7 heterodimer showed the inversion-specific band of the expected size (Fig. 3e). We conclude that a heterodimer of evolved recombinases can invert a 140 kb target sequence found in the human genome.

**Identification of potential human genomic off-targets**. The D7 heterodimer consists of two recombinases, which can form either a heterodimer or two different homodimers. Consequently, the amount of potential recognition sequences is increased and could result in the increased chance of unintended recombination at off-target sites. To identify potential off-targets, we bioinformatically analyzed the human genome for sequences with similarity to loxF8, loxF8-L, and loxF8-R (Supplementary Data 2)[17,25]. Thirteen predicted human off-target sites (Supplementary Fig. 11) were cloned into the pEVO reporter vector and tested in E. Coli. Eleven of these predicted human off-target-sites were not recombined by the D7 heterodimer, showing that D7 has high target-site selectivity. However, expression of the recombinase dimer revealed that one asymmetric (HG2) and one symmetric (HG2L) off-target sequence were indeed recombined by D7, albeit at lower efficiency than the loxF8 site (Fig. 4a).

**Linking of recombinase monomers reduces off-target recombination**. To reduce the chance of recombination at symmetric

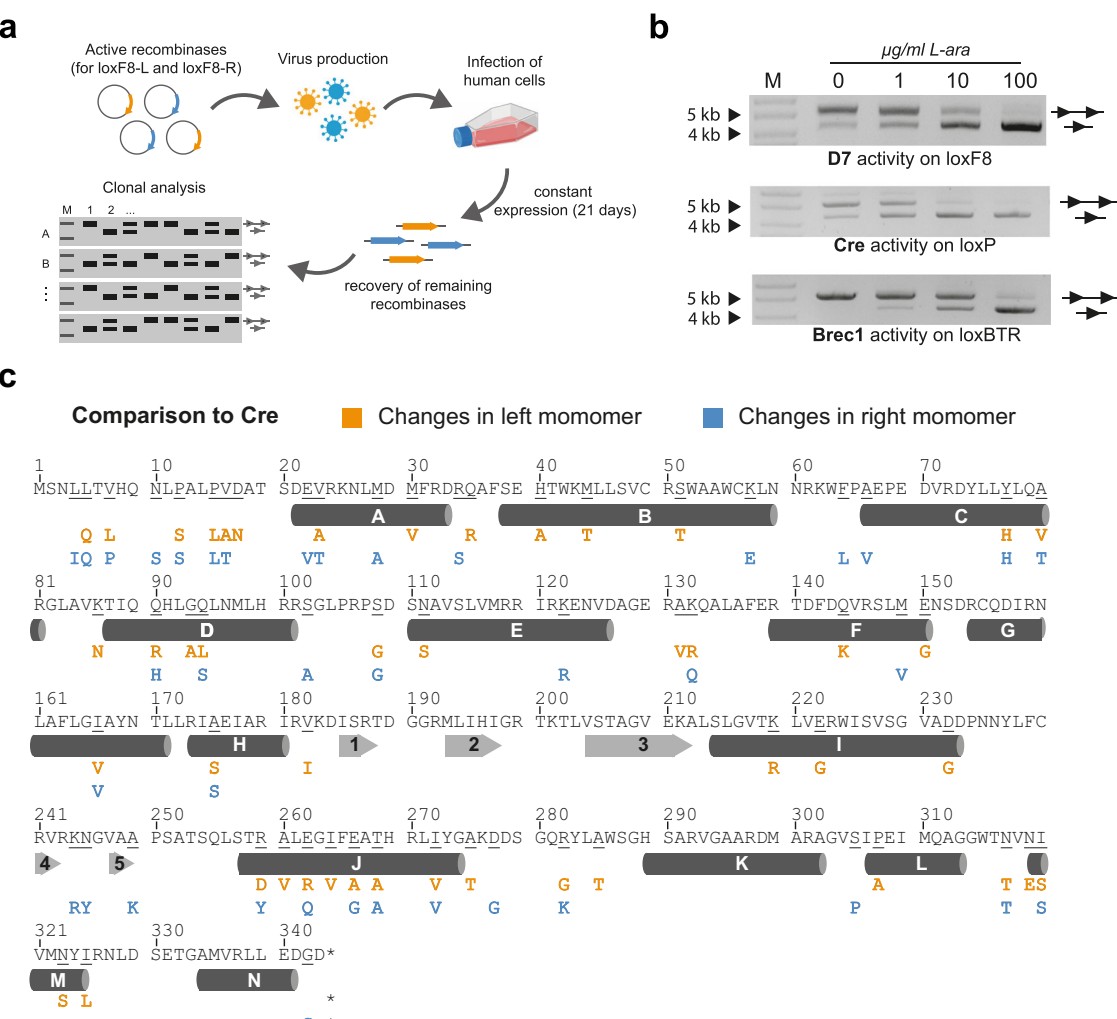

**Fig. 2 Identification of a loxF8 specific heterodimer clone. a** Overview of the clone selection procedure. Active recombinases for loxF8-L and loxF8-R are cloned to an inducible lentiviral expression plasmid. Human cells are infected with viral particles each coding for a recombinase from the active libraries. After 21 days of constant expression of the recombinases, genomic DNA of the cells is isolated and recombinase genes are retrieved via PCR. Recombinases are combined in a heterodimer and single clones are expressed in *E. coli*. Activity of the clones is analyzed using a PCR-based assay. The upper band represents the PCR product generated from non-recombined substrate (line with two triangles), whereas the lower band shows the PCR product generated from recombined substrate (line with one triangle). **b** Plasmid-based activity assay of the loxF8 candidate recombinase heterodimer (D7), Cre and Brec1. Recombinases were expressed in *E. coli* at four different L-arabinose concentrations (0, 1, 10, and 100 μg/ml). The upper band represents non-recombined substrate (line with two triangles), whereas the lower band shows the recombined plasmid (line with one triangle). The marker lane (M) for 4 kb and 5 kb is shown with arrows. **c** Mutation analysis of the D7 monomers (one-letter code). The amino acid sequence of Cre recombinase is shown as a reference. Orange and blue letters represent changes found in the left monomer and in the right monomer, respectively. Secondary structure elements are indicated, with alpha-helices displayed as cylinders with letters and beta-sheets represented as numbered arrows. Asterisks denote stop codons. Parts of the figure were created with BioRender.com. Source data are provided as a Source data file.

off-target sites, we focused on constraining the monomers from homodimerization. To achieve this goal, we physically fused the D7 monomers to enforce the desired heterodimer assembly, potentially abolishing recombination of symmetric off-targets by individual monomers (Fig. 4b). To fuse the monomers, we began with a common linker motif of eight glycine-glycine-serine (GGS) repeats[26]. Indeed, the fused D7 heterodimer recombined the loxF8 target site, although its activity was reduced compared to the non-fused D7 dimer, presumably because the linker prevented the individual monomers to assemble optimally on the DNA (Supplementary Fig. 12). In order to enhance the activity of the fused monomers, we designed a linker library that was screened for 10 cycles of alternating SLiDE to find the most active linker variants (Fig. 4c and Supplementary Fig. 12). Fused dimers were first selected for activity on the final asymmetric loxF8 site and

then subsequently for no activity on the symmetric loxF8-L and loxF8-R target sites (Fig. 4c). The final library showed robust activity on the loxF8 target site at low arabinose induction, indicating that variants with improved activity had been enriched (Supplementary Fig. 12). At this stage, twelve single clones were analyzed for their activity on loxF8, loxF8-L and loxF8-R. One clone (from here on referred to as RecF8) was selected, because it showed similar activity on the loxF8 target site as D7 at low induction levels (Supplementary Fig. 13). Noteworthy, RecF8 did not recombine loxF8-L and loxF8-R, thus abolishing activity on the symmetric target sites (Supplementary Fig. 13). Next, RecF8 was tested on the predicted human off-targets. In contrast to D7, RecF8 showed no observable activity on the symmetric HG2L off-target (Fig. 4d), signifying its improved properties by preventing homodimer formation of the individual monomers. Surprisingly,

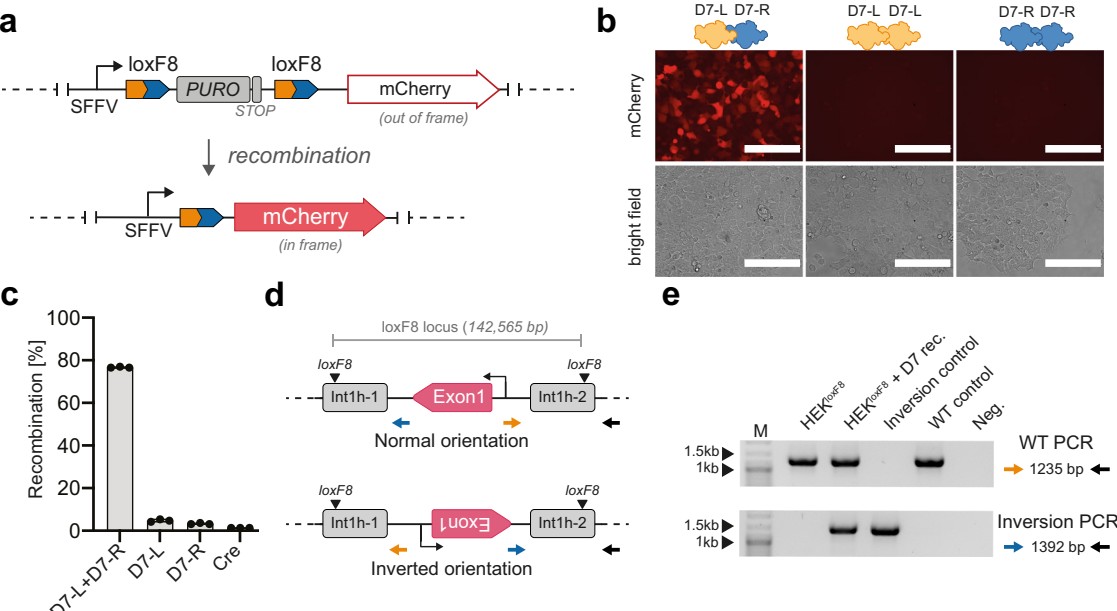

**Fig. 3 Activity of the D7 heterodimer in human cells. a** Schematic overview of the integrated reporter construct in HEK293T cells. Expression of mCherry is blocked by the puromycin (PURO) gene. After site-specific recombination the spleen focus-forming virus (SFFV) promoter-driven puromycin resistance cassette is excised, leading to expression of the red-fluorescent mCherry gene. **b** Fluorescent and brightfield images of transfected HEK293TloxF8 reporter cells with indicated recombinases. Note that mCherry expression is only visible in cells receiving both monomers. 200 µm scale bars are indicated. **c** Quantification of recombination efficiencies 48 h after transfection of HEK293T[loxF8] reporter cells with indicated recombinases, analyzed by flow cytometry ($n = 3$, biological replicates are shown as dots). Error bars represent standard deviation of the mean (SD). **d** Schematic overview of a fraction of the F8 gene displaying the PCR primers used to detect the orientation of the loxF8 locus. Exons are displayed in magenta and the repeated regions int1h-1 and int1h-2 are shown in gray. Primer binding sites are indicated with arrows. The position of the loxF8 target sites is indicated. The transcription start site of the F8 gene is depicted by a black arrow. **e** Gel image of PCR products generated using indicated primer combinations to detect the orientation of the loxF8 locus with and without treatment with the D7 heterodimer. Band sizes of the marker lane (M) are indicated. Inversion control = genomic DNA of patient-derived iPSCs carrying the exon1 inversion. WT control = genomic DNA of HEK293T[loxF8] reporter cells (non-treated). Neg. = water used as template for the PCR. Source data are provided as a Source data file.

the off-target activity on the asymmetric HG2 off-target site was also reduced (compare Fig. 4a, d, Supplementary Fig. 13), indicating that the linking might also have positive effects on asymmetric off-target activity.

Testing RecF8 in our HEK293T[loxF8] reporter cell line revealed that it is almost as active as the D7 monomers (Supplementary Fig. 14). More importantly, both RecF8 and D7 induce the inversion of the endogenous 140 kb loxF8 locus in ~30% of transfected cells after a single transient delivery of recombinase encoding mRNAs (Fig. 4e and Supplementary Fig. 14). Of note, based on the inversion equilibrium, the maximum possible inversion rate is 50%[11]. We conclude, that by fusing the two monomers to limit unwanted homotetramer formation, RecF8 exhibits improved specificity while retaining robust activity on the loxF8 on-target site in bacteria and human cells.

**An experimental off-target assay suggests high target site selectivity for RecF8.** Bioinformatic off-target predictions are limited to the depth of the input data consequently producing an incomplete off-target prediction profile for the desired recombinase. To increase the safety profile and enhance the classification of potential off-targets for designer-recombinases, we sought of an experimental method to nominate potential off-targets. Because DNA binding is a prerequisite for successful recombination, we developed a ChIP-Seq-based assay to identify putative RecF8 off-targets in the human genome (Supplementary Fig. 15). Eighty-five high confidence RecF8 binding sites were identified with this method (Supplementary Data 3). Twelve hits were subjected to validation by qPCR. For ten out of twelve

binding sites efficient primers could be designed and for nine out of these ten sites, in addition to the positive loxF8 control, we were able to confirm the RecF8-mediated DNA enrichment by qPCR (Fig. 4f). Hence, we were able to identify DNA regions, including the loxF8 on-target site (Supplementary Fig. 15) that are occupied by RecF8 in the human genome, thereby representing putative off-target sites. Local motif enrichment analysis revealed putative binding motifs within a 72 bp window around the peak summits (Supplementary Fig. 15). In order to test RecF8 activity on these potential off-target sites, we cloned 95 bp surrounding the peak summit as excision substrates into our bacterial and human reporter systems. Expression of RecF8 in bacteria and human cells did not result in any detectable recombination at these sites (Fig. 4g, Supplementary Fig. 15), indicating that these sequences, while bound, are not efficiently recombined by RecF8. We conclude, that ChIP-seq-based off-target profiling of RecF8 was able to identify potential binding sites in the human genome. These sites are most likely bound by the recombinase but not necessarily recombined.

Because D7 and RecF8 were both active on the HG2 off-target site in a plasmid-based assay in *E. coli*, we finally investigated whether this sequence is actually altered in human tissue culture cells upon RecF8 expression. Noteworthy, no HG2 recombination activity on the human sites could be detected with a PCR-based assay after expressing RecF8 at high levels (Supplementary Fig. 16), suggesting that RecF8-mediated alteration at this sequence does not occur in the human genome, at least not at frequencies detectable by PCR. We conclude that RecF8 is fairly specific with minimal detectable off-target activity.

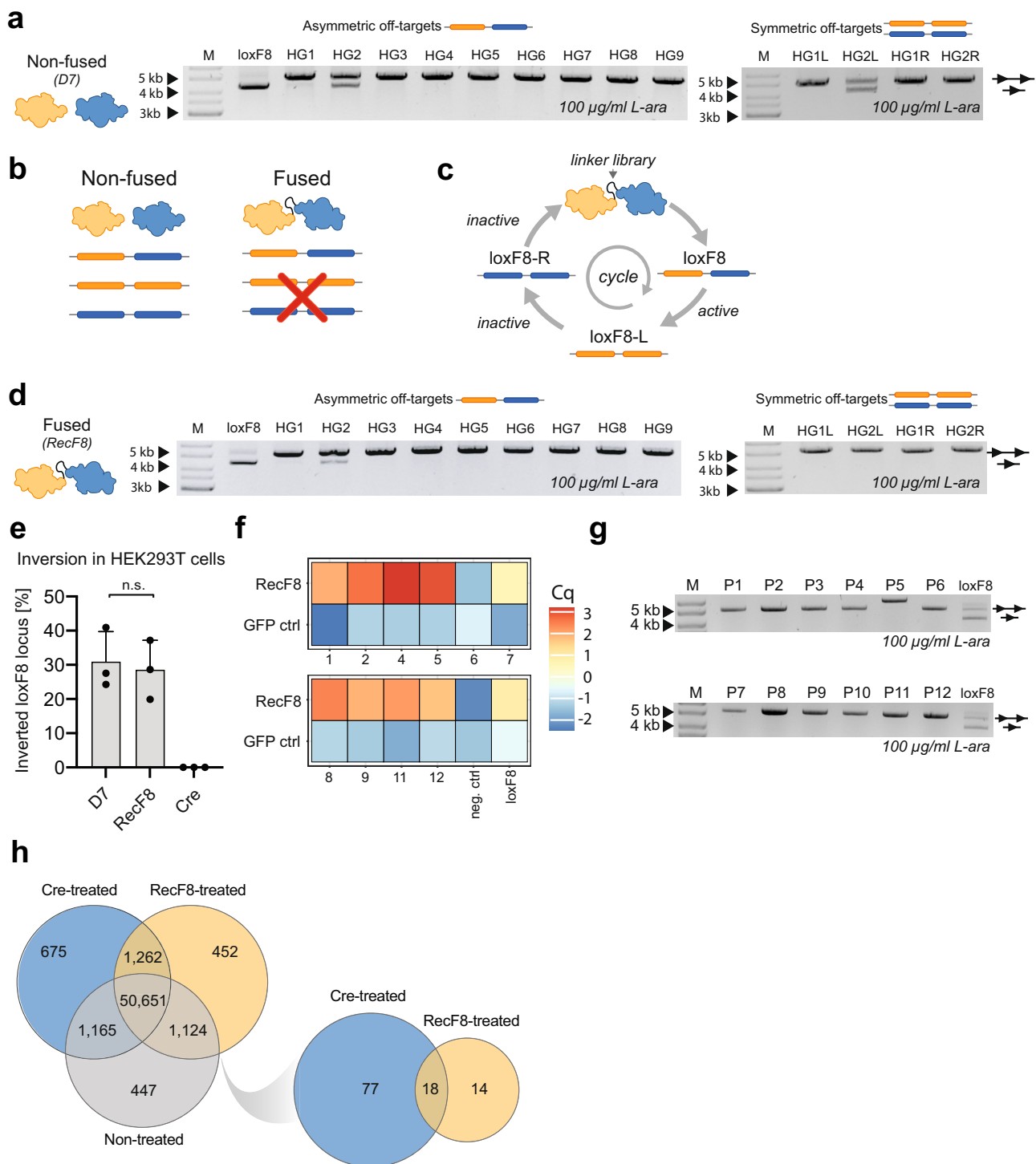

**Off-target analysis by long-read whole-genome sequencing**. In addition to the bioinformatic-based and ChIP-seq-based off-target profiling of RecF8 we performed whole-genome sequencing (WGS) as an unbiased approach of finding potential RecF8 induced off-target activity. To identify potential chromosomal aberrations, we employed Nanopore-based long-read sequencing technology. Three samples (Cre-treated, RecF8-treated and non-treated) derived from mRNA transfected patient-specific ECs were sequenced with a coverage of 35–40× each. A comparison to a reference genome revealed 50,651 potential common aberrations detected in all three samples, likely representing sequence differences of the patient to the reference genome. In contrast, 450–675 potential aberrations

were specific for the individual samples (Fig. 4h). The fact that the numbers of aberrations were almost identical for the non-treated sample (447) and the RecF8-treated sample (452) suggest that RecF8 expression does not cause extensive genomic rearrangements. Further analyses unmasked 109 aberrations specific to the recombinase-treated samples (Cre- and RecF8-treated). Of these 109 aberrations, 77 were specific for the Cre-treated sample, 18 were common and 14 were specific for the RecF8-treated sample, one of them being the successful int1h inversion (Supplementary Fig. 17). Hence, less putative off-target events were detected for RecF8 than for Cre, a recombinase that has been extensively used in human cells and in different animal models[11]. Most of the

**Fig. 4 Off-target analysis of F8 recombinases. a** Bacterial plasmid-based activity of the D7 heterodimer on predicted asymmetric and symmetric loxF8-like sites. The upper band represents non-recombined plasmid DNA (line with two triangles) whereas the lower band shows the recombined form of the plasmid (line with one triangle). The same applies for other gel pictures shown in this figure. **b** Schematic overview of physical linking of recombinases and the predicted activity on asymmetric and symmetric target sites. **c** Overview of the linker selection using alternating SLiDE. First, clones are selected for activity on loxF8. In the next two selection rounds these recombinases are selected against activity on symmetric target sites. Alternating the target sites and the selection of the linked recombinases allows for selection of clones with desired activity profiles (activity on loxF8, no activity on loxF8-L, and loxF8-R). **d** Bacterial plasmid-based activity of the linked RecF8 dimer on predicted human asymmetric and symmetric loxF8-like sites. **e** qPCR-based quantification of the genomic inversion efficiencies at the loxF8 locus after treatment with indicated recombinases in HEK293T cells ($n = 3$, biological replicates are shown as dots). Error bars represent standard deviation of the mean (SD). n.s. = not significant, unpaired two-sided $t$-test (two-stage step up method, Prism 8). **f** qPCR-based validation of putative binding sites identified by ChIP-seq. The RecF8 sample is compared to the GFP control sample. Enrichment is displayed by a heatmap of the difference of the Cq in the input sample and in the IP sample for each peak. **g** Bacterial plasmid-based RecF8 activity assay of the twelve RecF8-binding sites and on oxF8. The marker lane (M) for 4 kb and 5 kb is shown with arrows. **h** Venn diagrams showing counts of structural variants detected after whole-genome sequencing of recombinase-treated patient iPSCs, as well as in non-treated cells, in comparison to the reference genome. The diagram on the left represents counts of all called variants. The right diagram shows counts of deletions, translocations and inversions that are potentially caused by indicated recombinase activity. Source data are provided as a Source data file.

sequences around the potential RecF8 off-target hits were repetitive sequences and of low GC-content, which made it difficult to design primer pairs for validation experiments. Nevertheless, PCR validation of one off-target locus revealed that the putative RecF8-mediated deletion was also detectable in the non-treated sample (Supplementary Fig. 17), indicating that this aberration was not caused by RecF8 expression and likely represents a false-positive result.

Altogether, data from bioinformatically predicted off-targets, ChIP-seq-based off-target profiling, and WGS leads us to conclude that RecF8 is fairly specific, without causing substantial unwanted genome alterations.

**RecF8 expression corrects the F8 exon 1 inversion in patient-derived iPSCs.** In order to investigate whether RecF8 can correct the disease-causing exon 1 inversion in patient-derived cells, iPSCs were generated from blood cells of a hemophiliac donor (Fig. 5a). The iPSCs were characterized in detail and protocols for efficient mRNA transfections were established (Supplementary Fig. 18). Seventy-two hours post transfection with RecF8 mRNA, genomic DNA was isolated and analyzed for the inversion of the loxF8 locus via PCR. Only patient-specific iPSCs (F8 iPSCs) treated with RecF8 showed the expected PCR bands, indicating that the loxF8 locus had been inverted (Fig. 5b). Sequencing of the PCR bands revealed that the inversion in the F8 iPSCs resulted in the same sequence as it was found in non-treated WT iPSCs. This result demonstrates that the RecF8-mediated inversion brings exon 1 of the F8 gene back into the right orientation and thereby corrects the genetic defect in patient-derived cells at nucleotide precision. Conversely, the same sequence found in patient-derived iPSCs was identified in WT iPSCs after treating them with RecF8 (Fig. 5c). This result validates that the genomic inversion of the 140 kb endogenous loxF8 locus can be induced in human iPSCs in both directions.

**Factor VIII expression is restored in RecF8-treated patient-derived endothelial cells.** It has been suggested that endothelial cells (ECs) of the liver are the main producer cells for Factor VIII in the human body[27,28]. In order to test if F8 gene expression can be restored in this cell type in vitro, we differentiated patient-derived iPSCs into endothelial cells (ECs) and delivered RecF8 in form of mRNA into the cells (Supplementary Fig. 19). Seventy-two hours post transfection, isolated genomic DNA was analyzed for the inversion of the 140 kb endogenous loxF8 locus using a PCR-based assay (Fig. 6a). Only the patient-derived ECs treated with RecF8 mRNA showed the PCR bands specific for the WT control (Fig. 6a). Quantification of the genomic DNA inversion in

ECs treated with the RecF8 mRNA revealed that ~12% of the loxF8 locus had been reverted back to the WT orientation (Fig. 6b). To examine whether the corrected cells express Factor VIII, we isolated mRNA from non-treated and RecF8-treated cells and performed qRT-PCR assays. No amplicon could be generated from the untreated ECs, consistent with the inability of these cells to produce Factor VIII mRNA. In contrast, up to ~6% of the Factor VIII transcript could be detected after treatment of ECs with RecF8 mRNA (Fig. 6c). Moreover, sequencing of the amplified cDNA fragment revealed that the transcript showed the right boundary of exon 1 and exon 2 (Fig. 6d) and sequencing of other exons, as well as immunostainings, indicated the expression of the full-length Factor VIII transcript and protein (Supplementary Fig. 19). These results demonstrate that the evolved RecF8 recombinase can be used to specifically invert the 140 kb DNA fragment between int1h-1 and int1h-2 and restore the expression of the F8 gene in patient-derived ECs.

It has been shown that F8 gene inversions can be corrected using nucleases such as TALENs or CRISR/Cas9[6,7]. Reported efficiencies of up to 6,7% inversion rates are in line with our RecF8-mediated results. However, the nuclease-mediated inversions are a result of the cellular DNA repair after cutting the DNA. In contrast, recombinases, such as RecF8, modify the DNA independent of intrinsic DNA repair processes. Therefore, we were interested in the fidelity of RecF8 compared to a CRISPR-based inversion of the loxF8 locus. To compare the two methods in this regard, patient-specific ECs were treated with either RecF8 or Cas9 mRNAs in combination with a gRNA specific for the inverted repeat. The inversion products detected by PCR were then subjected to sequencing (Supplementary Fig. 20). For each condition, 22 clones were analyzed. No sequence alteration was found for the RecF8 treated sample, demonstrating the precision of the recombinase-mediated correction. In contrast, only three sequencing reads from the CRISPR-based inversion showed no alterations (Fig. 6e and Supplementary Fig. 20). Conversely, 19 reads showed Indels of different sizes. In fact, half of the sequenced clones showed deletions of more than 100 base pairs (Fig. 6e and Supplementary Fig. 20), substantially changing the genomic sequence at the F8 locus. These results confirm the fidelity of RecF8, demonstrating that designer-recombinases have considerable potential for high fidelity correction of large gene alterations, especially genomic inversions.

## Discussion

Genome editing tools are rapidly being developed for therapeutic applications to devise novel curative strategies for monogenic diseases[29]. Gene inversions are a particularly challenging genetic alteration to correct with the most commonly used nuclease-based genome editing tools[6,7]. We explored the potential of site-specific

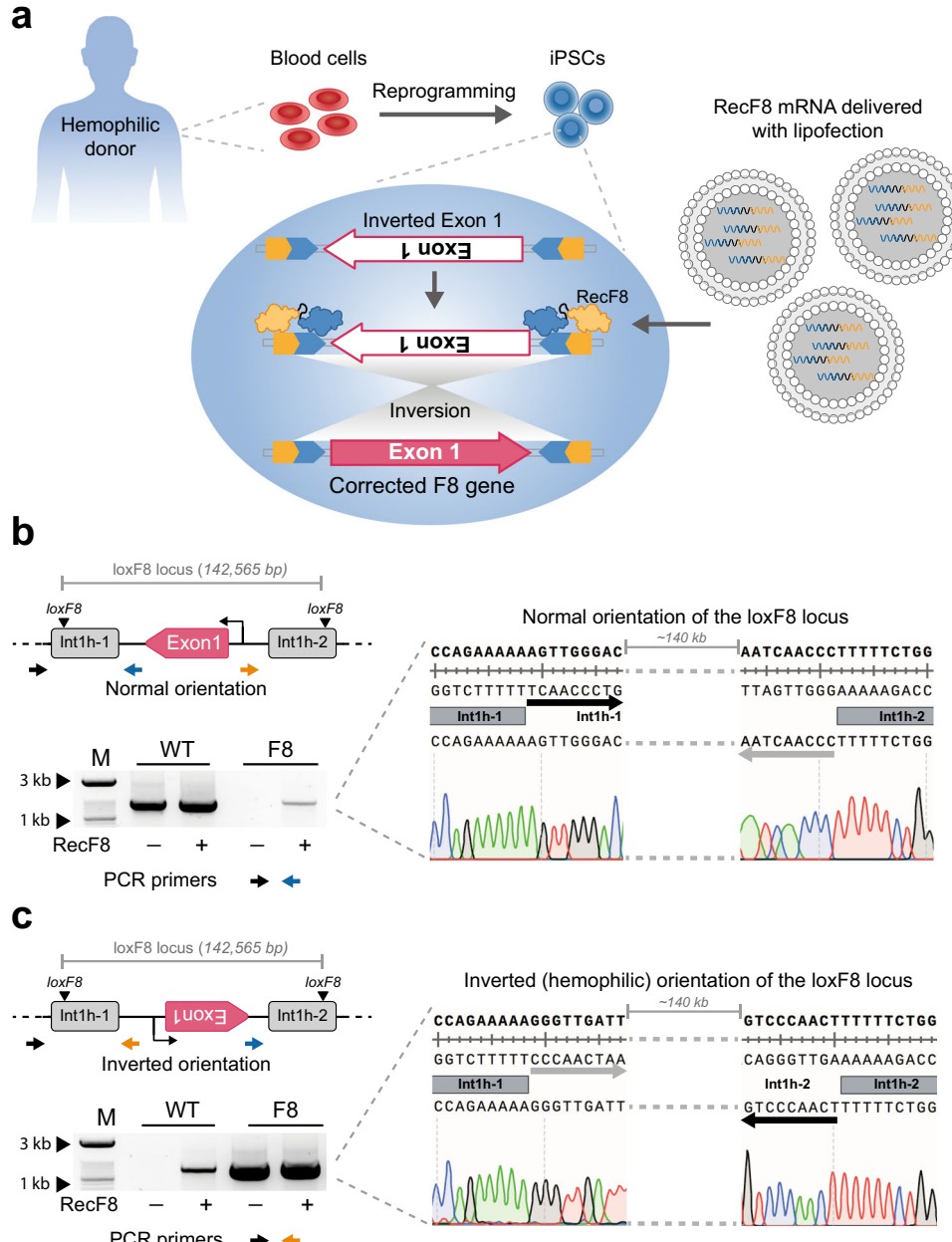

**Fig. 5 Correction of exon 1 inversion in patient-specific cells. a** Schematic representation of the generation of patient-specific induced pluripotent stem cells (iPSCs) and treatment with RecF8 mRNA. A magnification of the nucleus is shown with a schematic depiction of the loxF8 locus before and after treatment with RecF8. The two loxF8 target sites are shown in yellow and blue. **b** PCR-based detection of the loxF8 locus in WT and patient-derived iPSCs with and without RecF8 treatment. Employed primers are depicted in the illustration by arrows in black, blue and orange. **c** PCR-based detection of inverted loxF8 locus in WT and patient-derived iPSCs with and without RecF8 treatment. Sequencing reads of the PCR products confirming the corrected orientation of the DNA sequence after RecF8 treatment are displayed. The black and gray arrows indicate the sequence orientation between the int1h-1 and int1h-2 repeats of the loxF8 locus. Marker (M) lanes at 1 kb and 3 kb are indicated. WT = iPSCs from a donor that does not carry the exon 1 inversion of the F8 gene. F8 = iPSCs from a hemophilic donor carrying the exon 1 inversion of the F8 gene. Source data are provided as a Source data file.

recombinases (SSRs) to correct a large gene inversion frequently found in Hemophilia A. SSRs have previously been shown to catalyze large genomic inversions and their target specificity can be altered through directed molecular evolution[14,15,30,31]. Another advantage of SSRs is their independence of the host cell DNA repair machinery and other cofactors. This allows these enzymes to work efficiently in vivo in most cell types, including post-mitotic cells[11].

To obtain SSRs that recognize a sequence in the inverted int1h repeats surrounding the F8 gene, we applied the well-established

substrate-linked directed evolution approach[14,16,17,21] and obtained Cre-type enzymes with activity on the desired sites. We acquired both single and heterodimeric Cre-type recombinase systems with robust activity on the loxF8 target sites, demonstrating multiple strategies to successfully bypass the native symmetric preference of Cre. The heterodimeric system required less evolution cycles to develop a novel recombinase of comparable efficiency, suggesting that this method might be a more economical approach for asymmetric target sites. Additionally, developing a heterodimeric recombinase system consisting of evolved monomers with different

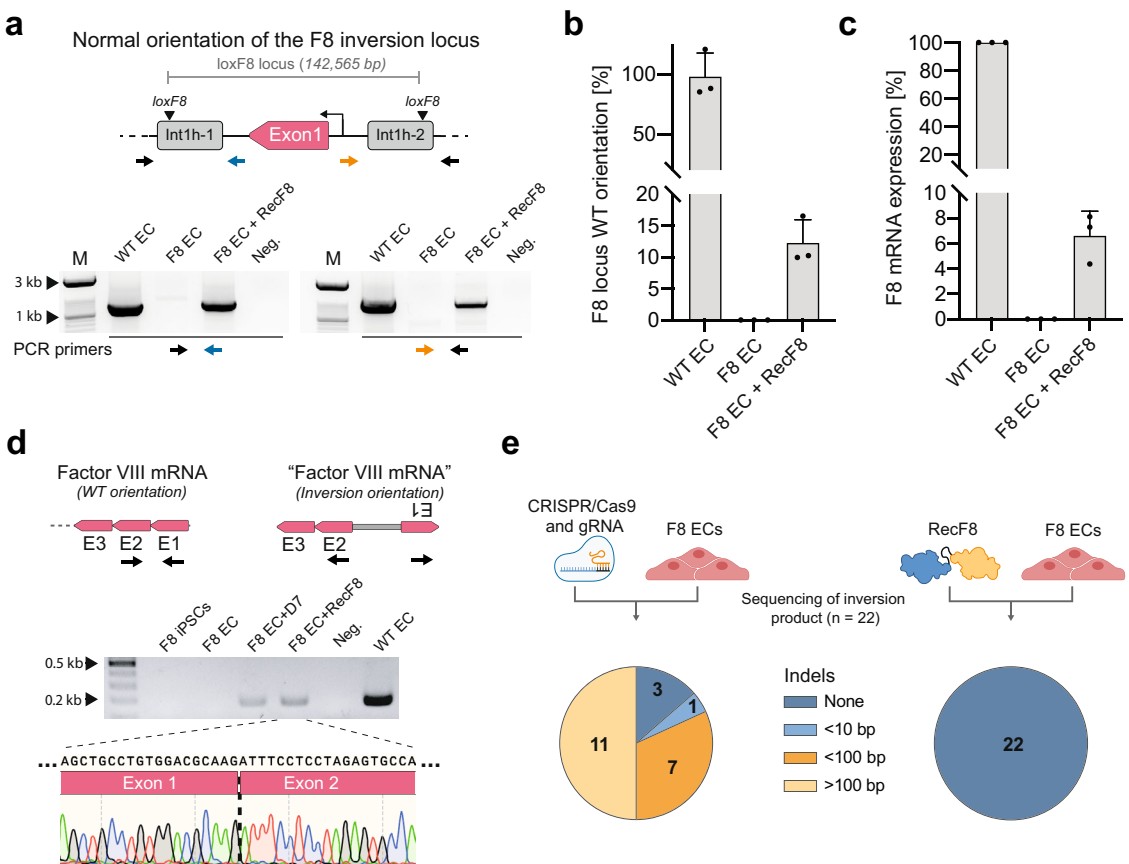

**Fig. 6 Functional correction of patient-specific endothelial cells differentiated from iPSCs. a** RecF8 expression corrects the int1h inversion. Overview of primer arrangement around the first and second loxF8 site to detect the orientation of the 140 kb fragment (top panel). Agarose gel of PCR products obtained on genomic DNA from patient-specific ECs with or without treatment with RecF8 (lower panel). Marker (M) lanes at 1 kb and 3 kb are indicated. WT EC = ECs differentiated from a donor that does not carry the exon 1 inversion. F8 EC = ECs differentiated from a hemophilic donor carrying the exon 1 inversion. '+RecF8' = cells were transfected with RecF8 mRNA. Neg. = water control. **b** qPCR-based quantification of the genomic inversion efficiencies in patient-specific ECs with and without RecF8 treatment. Inversion quantification of WT ECs is shown as reference (n = 3, replicates are shown as dots). Error bars represent standard deviation of the mean (SD). **c** qPCR-based quantification of Factor VIII mRNA transcript. The relative expression of the Factor VIII mRNA was measured after treating patient-specific ECs with or without RecF8 mRNA (n = 3, replicates are shown as dots). Error bars represent standard deviation of the mean (SD). Factor VIII mRNA expression in untreated WT ECs was used for normalization. **d** Schematic depiction of the first three exons of the Factor VIII mRNA and the primers used for detecting the transcript. A gel image and sequencing read of Factor VIII cDNA of RecF8 treated patient-derived ECs is shown. The exon1–exon2 boundary can only occur, if the inversion is corrected to the WT orientation, the gene is transcribed and the pre-mRNA spliced correctly. **e** Quantification of indels introduced by CRISPR/Cas9 or RecF8 after inverting the loxF8 genomic locus. Counts of clones harboring different kinds of indels are shown in the pie charts. 22 clones were analyzed for each condition. Parts of the figure were created with BioRender.com. Source data are provided as a Source data file.

sequence specificities provides greater flexibility when selecting potential target sites because the asymmetries in each half-site are no longer a limitation[21]. However, using two evolved monomers instead of one increases potential off-target activity. Each monomer can be combined into a homotetrameric complex to recombine potential symmetric off-targets or into a heterotetrametric complex to recombine asymmetric off-targets. We mitigated this problem by linking the two monomers forming a heterodimer (RecF8). The fusion of the two monomers eliminated observed off-target recombination of a symmetric target site and reduced the effect on an asymmetric off-target site that the non-linked recombinase actively recombined. We assume that the linking of the two monomers sterically obstructed synapse formation and allowed only activity on asymmetric target sites where the linked monomers could bind simultaneously. We believe that this approach should be generally applicable and expedite the process to generate novel, evolved hetero-specific recombinases with desirable properties in the future.

To experimentally identify potential recombinase off-targets, we present a ChIP-based approach. The assay works well to identify recombinase binding-sites in the human genome. However, DNA binding did not necessarily correlate with active recombination-target sites. Possibly, combining the ChIP-based approach with data from bioinformatic predictions and WGS could improve tools to identify putative off-target signatures of designer-recombinases.

Single-dose mRNA-mediated expression of RecF8 resulted in up to 30% inversion of the loxF8 locus in HEK293T cells and in up to 12% in patient-specific iPSC-derived endothelial cells. As the limit for SSR-mediated inversion is 50%[11], these results demonstrate the potential of SSR as an inversion genome editing tool. The genomic inversion in patient-derived cells led to 6% F8 mRNA expression compared to control cells, providing a solid starting point to explore this approach further.

Nevertheless, conclusive results of how much inversion of the F8 gene can ultimately be reached need to be evaluated in an F8

inversion animal model. Ideally, in such an animal model, the recombinase would be transiently delivered to the cells of interest for example, with the use of lipid nanoparticles packaged mRNA or through adeno-associated virus (AAV)-mediated delivery[32,33]. These studies should also reveal if RecF8 expression is tolerated and does not cause unwanted side effects. If necessary, RecF8 specificity could be further improved utilizing established methods[34–37].

Overall, our work introduces designer-recombinases as a promising genome-editing tool to correct human diseases caused by genomic inversions. Importantly, numerous other human diseases have already been identified to be caused by genomic inversions, including the intron 22 inversion in Hemophilia A[20], Sotos Syndrome[38], or Hunter Syndrome[39], with additional disease-causing inversions likely waiting to be discovered[40,41]. All of these could potentially benefit from a genome editing strategy employing designer-recombinases.

## Methods
Primer lists and cycling programs can be found in Supplementary Data 4 and Supplementary Table 1.

**Target site identification**. In order to nominate a suitable target site, we first identified a sequence of the inverted repeat int1h, by aligning the first intron of the F8 gene against a reverse-complement sequence of 200 kb DNA fragment located upstream of the F8 transcription start site (reference genome assembly hg38), using EMBOSS Water as an alignment tool[42]. In the following step, using a python script, the inverted repeat (1041 bp) was scanned for all occurrences of potential target sites, defined as palindromic repeats of 13 bp (half-sites) separated by 8 bp spacer sequence. The half-sites were allowed to have up to 7 positions of asymmetry. A set of 82 target sites fulfilling this criterion was sorted by a larger count of mismatches between one of the half-sites (left or right) and any of the half-sites recognized by previously evolved Cre-type recombinase libraries[16,17]. As the final target site, we picked the sequence with the highest score from the sorted list and named it loxF8.

In order to avoid potential off-targeting, we scanned the human genome for any occurrences of the selected target site, allowing up to one mismatch in each half-site and any sequence in the spacer region. A search for genomic sequences with the highest resemblance toward loxF8 was performed using an exhaustive short sequence aligner PatMaN, allowing up to 8 mismatches in both half-sites in total[43].

**Plasmid construction**. Evolution vectors with the different target sites for SLiDE were cloned as described previously[21]. In short, primers were designed to carry the desired target sites and an overlap with the pEVO vector. The PCR fragment generated when using the pEVO vector as template was then cloned via Cold Fusion into a BglII digested pEVO backbone (System Biosciences).

A DNA fragment (synthesized by Twist Bioscience) coding for TRE3G-EF1a-Tet-ON® 3G-P2A-eGFP was inserted into the lentiviral backbone of the plentiSAMv2, a gift from Feng Zhang, (Addgene plasmid #75112; http://n2t.net/addgene:75112; RRID:Addgene_75112)[44] utilizing NheI and KpnI restriction enzymes (NEB). The resulting plasmid (plentiX) can be used to clone recombinases under the control of a doxycycline inducible promoter employing BsrGI and XbaI restriction enzymes (NEB). In order to fuse the recombinases to EGFP the plasmid was modified and a TRE3G-NLS-eGFP-EF1a-Tet-ON®3G-P2A-PURO cassette (synthesized by Twist Bioscience) was inserted between the LTRs. The resulting plasmid was used to exchange EGFP with mCherry or tagBFP.

The plentiCRISPR v2, a gift from Feng Zhang, (Addgene plasmid #52961; http://n2t.net/addgene:52961; RRID:Addgene_52961)[45] was used to generate the loxF8 reporter plasmid containing a SFFVpromoter-loxF8-PURO-loxF8-mCherry cassette between the LTRs.

**Cloning and expression of recombinases**. Single recombinases are cloned into the pEVO vector utilizing BsrGI and XbaI restriction enzymes (NEB). Dimer recombinases were cloned into the pEVO vector employing BsrGI and XbaI (second position) and SacI and XhoI restriction enzymes (first position). A Shine-Dalgarno (SD) sequence was located in front of each recombinase gene. In the pEVO vector carrying two recombinases, the SD sequence allows bicistronic expression of both recombinases. Expression of recombinases was controlled by an L-arabinose inducible promoter system (araBAD).

One or two recombinases were cloned into the mammalian plentiX recombinase expression vector under the control of a doxycycline inducible promoter. The first recombinase is cloned employing BsrGI and XbaI restriction enzymes (NEB). If the expression of two recombinases is intended, the stop codon of the first recombinase is removed and the second recombinase is cloned utilizing EcoRI and XhoI restriction enzymes. The plentiX vector harbors a SV40 NLS

before each recombinase cloning site. A T2A site situated between the first and second recombinase gene enforced expression of both recombinases.

Transient expression of recombinases was achieved by cloning the recombinase gene in a mammalian expression vector (EF1a-Rec-P2A-EGFP) utilizing BsrGI and XbaI restriction enzymes (NEB). The STOP codon of the recombinase was removed to allow translational linking with EGFP using a P2A self-cleaving peptide sequence. Recombinase expression was driven by an EF1a promoter.

**Substrate-linked directed evolution (SLiDE)**. Recombinases were evolved using substrate-linked directed evolution as described previously[14,16,17,21]. A schematic overview of the experimental procedure is depicted in Fig. 1b. Recombinase libraries were evolved stepwise on different subsites to obtain active variants on the final target sites loxF8, loxF8-1, loxF8-2, loxF8-L, and loxF8-R.

In short, two parallel evolutions were set up to evolve either a recombinase monomer or a recombinase dimer being able to recombine the final asymmetric loxF8 target site. In the first step different pEVO vectors were generated containing the evolution sites (Supplementary Figs. 2 and 3) as a recombinase excision substrate. Diversified Cre-type recombinase libraries from previous evolutions were used to initiate the evolution process[14,16,17,21]. These libraries were cloned via XbaI and BsrGI in the pEVO vectors containing the first subsites (loxF8-1a, loxF8-2a, loxF8-L2 and loxF8-R2) and recombinase expression was induced at high levels (200 μg/ml L-arabinose). Recombinases active on the target sites will excise a part of the pEVO vector containing two unique restrictions sites (AvrII and NdeI) making this vector 'immune' to a subsequent digest with AvrII and NdeI-HF (NEB). These recombines were then retrieved using a PCR designed to only amplify from a circular template (Fig. 1b). This PCR step was performed with a low-fidelity DNA polymerase (MyTaq, Bioline, primers 79 + 80) to incorporate random mutations in the recombinase gene. The amplified recombinases were then cloned into a non-recombined pEVO vector and recombinase expression was induced again. This cycling process was repeated with lowering the recombinase expression by reducing the L-arabinose concentration (from 200 μg/ml down to 1 μg/ml) to select for the most active recombinase variants. SLiDE evolution was stepwise guided through the different evolution-target sites to obtain either an active monomer or an active dimer library for the final loxF8 target site (Supplementary Figs. 2 and 3).

**Clonal analysis of recombinases**. Recombinase activity was either analyzed with a plasmid-based assay or a PCR-based assay as previously described[21]. Schematics of both assays are depicted in Fig. 1b and Supplementary Fig. 6a. Briefly, recombination of the respective target sites on the evolution plasmid leads to the excision of a fragment from the plasmid. The resulting size difference is an indication for recombinase activity and can be detected by a restriction digest or by PCR. In order to detect the recombination by PCR a three-primer assay was used. The amplicon derived using primers 1 + 2 (Supplementary Fig. 6) detects the non-recombined product as the primer 2 binds in between the lox-sites. The resulting PCR product is 475 bp. Another primer set (1 + 3, Supplementary Fig. 6) will generate an amplicon (400 bp) from the recombined plasmid. The elongation time of the PCR was chosen so that primer 1 + 3 will not generate any amplicon on non-recombined templates.

**Recombination quantification**. Recombination efficiency of candidate recombinases was quantified using a plasmid-based assay. The recombined pEVO is smaller in size compared to the non-recombined version. After linearization, this size difference can be visualized on a standard agarose gel by gel electrophoresis. The bigger band (~5 kb) shows non-recombined substrate, while the smaller band (~4.3 kb) shows recombined substrate. Recombination efficiencies were calculated by measuring the bands intensities using Fiji. The average recombination efficiency was calculated from gel images of three independent biological experiments.

**Cell culture of HEK293T and HeLa cells**. HEK293T (ATCC) and HeLa (MPI-CBG, Dresden) were cultured in DMEM, Dulbecco's modified Eagle's medium (Gibco) with 10% fetal bovine (tetracycline-free) serum and 1% Penicillin- Streptomycin (10,000 U/ml, ThermoFisher).

**Fluorescent activated cell analysis**. HEK293T, HeLa, or iPSCs were washed once with PBS and then detached using Trypsin (Gibco) or Accutase (Sigma). Cells were resuspended in FACS buffer (PBS with 2.5 mM EDTA and 1% BSA) and analyzed with the MACSQuant® VYB Flow Cytometer (Miltenyi) or the BD FACSCanto™ II Cell Analyzer (BD). Analysis of the data was performed using FlowJo™ 10 (BD). Gating strategies are displayed in Supplementary Fig. 21.

**Long term expression of recombinase libraries**. Recombinase libraries were cut out of the evolution vectors via BsrGI and XbaI restriction sites, purified from an agarose gel and subsequently ligated to the plentiX viral vector. HeLa (MPI-CBG Dresden) cells were transduced with lentiviral particles generated from the lentiX-loxF8-L and lentiX-loxF8-R libraries. Transduced HeLa cells will express EGFP and recombinase expression can be induced upon administration of doxycycline. Cells were transduced with a MOI of one to enrich for clones harboring only one

recombinase integration per cell. Recombinase expression was induced for 21 days with 100 ng/ml doxyclcline and the medium was renewed every other day. Next, genomic DNA was isolated using the QIAamp DNA Blood Mini Kit (Qiagen) and recombinases were retrieved by PCR using the high-fidelity Herculase II Phusion DNA polymerase (Agilent). Retrieved recombinase libraries were ligated into the evolution vectors and the activity of single clones and libraries was assessed by PCR or by a digest as previously described earlier.

**Generation of the HEK293T^{loxF8} and HEK293T^{loxP} cell lines**. HEK293T (ATCC) cells were transduced with lentiviral particles generated from the SFFVpromoter-loxF8-PURO-loxF8-mCherry or SFFVpromoter-loxP-PURO-loxP-mCherry plasmids. The cells were exposed to 2 μg/ml puromycin selection 48 h after transduction for 7 days. Different concentrations of viral particles were used for transduction to estimate the MOI. The viral load that resulted in a transduction efficiency of ~5% (95% of cells were depleted during the puromycin selection) was used to establish reporter cell lines with an estimated MOI of 1. Genomic DNA was isolated from the surviving cells and a reporter-specific PCR was performed. The PCR fragment was sequenced to confirm that the reporter construct was integrated in the genome.

**In vitro transcription**. Recombinase, eGFP, tagBFP, and mCherry mRNA was produced by in vitro transcription (IVT) using the HiScribe^{TM} T7 ARCA mRNA Kit (NEB) and purified using the Monarch© RNA Cleanup Kit (NEB). The DNA templates for the IVT were generated by PCR using the lentiviral plasmids with eGFP (primers 1 + 2), mCherry (primers 3 + 4), tagBFP (primers 5 + 6). Recombinase DNA templates for IVT were generated by PCR using the pEVO vectors as templates with D7 (primers 7 + 8 and 9 + 10) or RecF8 (primers 11 + 12). Cycling program 3 was used for the mRNA transcription and poly(A) tailing reaction.

**mRNA transfection**. IVT produced mRNA was transfected using Lipofectamine™ MessengerMAX™ Transfection Reagent (ThermoFisher). HEK293T^{loxF8} cells were transfected in a 12-well format, seeded at a density of 250,000 cells/well and transfected 24 h after seeding. For each well 250 ng of mRNA (200 ng recombinase mRNA and 50 ng tagBFP mRNA) and 2 μl Lipofectamine™ MessengerMAX™ was mixed with 50 μl Opti-MEM I Reduced Serum Medium (ThermoFisher, pre-warmed at RT). After 5 min of incubation at RT the mRNA and Lipofectamine™ MessengerMAX™ mixtures were combined, shortly vortexed and incubated for 10 min. The transfection mixture was directly added to the cells without changing the medium. Cells were analyzed 48 h post transfection by FACS and fluorescent microscopy.

iPSCs were cultured using StemFit basic 02 (AJINOMOTO) and iMatrix-511 silk laminin coating (NIPPI) as described elsewhere[46]. In short, iPSCs were maintained in 6-well plates using StemFit 02 (Reprocell/AJINOMOTO) supplemented with 100 ng/ml FGF2 (R&D) and iMatrix-511 silk laminin coating (NIPPI). 10 μl laminin in 2 ml PBS for each well was used for coating and incubated 1 h at 37 °C. Confluent iPSCs were washed once with 2 ml PBS and detached using 1 ml Accutase (ThermoFisher) for 1 min at 37 °C. Cells were collected in 10 ml DMEM F12 (ThermoFisher) and pelleted for 3 min at 300 × g. The pellet was dissolved in 5 ml DMEM F12. Depending on the cell line 1 × 10^5–5 × 10^5 cells/well were plated on coated plates. The first 24 h after splitting the medium was supplemented with 10 uM Rock-inhibitor (Y-27632, Tocris). Medium change was performed daily. iPSCs were transfected as previously described[47].

**Detection of the int1h inversion by PCR on genomic DNA**. Genomic DNA of HEK293T^{loxF8}, iPSCs and ECs was isolated 48 h post transfection using the QIAamp DNA Blood Mini Kit (Qiagen). Two sets of primers were designed to detect the orientation of the 140 kb DNA fragment between the two loxF8 target sites. Primers (15 and 18) are located outside the fragment and do not change upon recombination. Primers (16 and 17) bind inside the 140 kb fragment and will change their orientation upon recombination. Primer pairs 15 + 16 and 17 + 18 were used to detect the WT orientation of the 140 kb fragment. Primer pairs 15 + 17 and 16 + 18 were used to detect the inverted variant. Independent of the orientation and the primer combinations, cycling program 1 was used for the PCR.

**Linker selection**. The linkers of eight GGS repeats were synthesized by Sigma-Aldrich as annealing compatible oligonucleotides. The designed linker library contained the core 12 amino acids coded by the degenerate codon RVM that were flanked by two GGS repeats from each side, and the sequence was synthesized. In both cases, the linker sequences were inserted in the pEVO plasmid via XhoI and BsrGI, between the two recombinase monomers. The linker selection was performed following the SLiDE protocol, with the exception of using the high-fidelity Herculase II Phusion DNA polymerase (Agilent) in order to select the heterodimer fused with a linker with the best properties without introducing new mutations in the recombinase sequences. By varying selection of active and inactive recombinase heterodimers on the loxF8 and symmetric sites (loxF8-L and loxF8-R), respectively, the counter selection was performed. Active recombinases on the loxF8 site were retrieved the same way as described previously in the SLiDE procedure. Inactive

variants for loxF8-L and loxF8-R were retrieved by PCR using a primer binding between two recombinase target sites and another primer binding before the recombinase dimer (primers 79 + 81). Only recombinase coding sequences carrying an inactive recombinase for loxF8-L or loxF8-R will be amplified. This simple assay selection of linked heterodimers that are able to recombine the final asymmetric loxF8 site, but do not recombine the symmetric loxF8-L and loxF8-R target sites.

**Inversion quantification**. Quantification of the inversion efficiencies was performed with a qPCR-based assay. In order to detect the WT orientation primer pair 17 + 18 was used together with a TaqMan specific probe for this amplicon. To detect the inversion orientation primer pair 16 + 18 was used. For both reactions cycling program 2 was used. A standard curve of 1%, 5%, 10%, 25%, 50%, and 100% inversion was generated by mixing genomic DNA of WT iPSCs and F8 iPSCs at appropriate rations. The standard curve was used to extrapolate the inversion efficiency of the genomic DNA samples. As the standard curve was generated using genomic DNA of male iPSCs (one X-chromosome), the calculated inversion efficiencies for the HEK293T cells (female, two X-chromosomes) were divided by two.

**ChIP-Seq and qPCR validation of potential binding sites for RecF8**. RecF8 recombinase was fused with EGFP and cloned in a modified version of the tetracycline-inducible plentiX vector. Hela cells were infected with the lentivirus and selected with 2 μg/ml puromycin 48 h after transduction for 7 days.

Cells were grown in 10 cm dishes and the expression of RecF8 or EGFP (control) was induced for 24 h with 100 ng/mL doxycycline. The cell lines were crosslinked with 1% formaldehyde for 10 min at room temperature and further processed following the manufacturer's protocol for High Cell number using the kit TruChip Chromatin Shearing Kit (Covaris). Chromatin shearing was performed using a Covaris M220 sonicator. 1% of the sheared chromatin was separated for qPCR validation (input sample) and the rest was used for immunoprecipitation.

Sonicated chromatin was immunoprecipitated using a goat GFP-antibody (MPI-CBG antibody facility, 1:5000) and Protein G sepharose beads (Protein G Sepharose® 4 Fast Flow, GE Healthcare). Eluates were reverse crosslinked followed by RNA and protein digestion.

Sequencing libraries were prepared using NEBNext® Ultra™ DNA Library Prep Kit for Illumina® from 17 to 75 ng of ChIP DNA with 15 PCR amplification cycle and size selection using AMPure XP beads. Paired-end sequencing was performed on an Illumina HiSeq 2000, aiming for approximately 25 million pairs of sequencing reads per sample, with each read being 76 bp long. Sequencing reads were aligned to a human reference genome assembly GRCh38.p12[48] using STAR aligner[49] tuned for ChIP-Seq analysis pipeline (disabled intron detection, 400 bp of maximum gap between read mates, up to 50 reported alignments of multi-mapper reads). Peak calling was performed with Genrich (https://github.com/jsh58/Genrich), using the ENCODE blacklist (v2)[50] for filtering out problematic regions. All steps involving manipulations and comparisons of genomic intervals were done using BEDTools[51]. Visualizations of the ChIP-Seq pile-up signals were generated with the USCS Genome Browser[52] (USCS genome browser tracks are available upon request), directly from BAM files sorted by samtools command line tool[53].

Ten out of twelve peaks that were identified by ChIP-Seq were additionally tested by qPCR for recombinase binding (Primers 49-78). qPCR analysis comparing immunoprecipitated samples and input samples was performed using SYBR green MasterMix (Thermo Scientific ABsolute qPCR SYBR Green Mix).

De-novo motif discovery was performed with MEME-ChIP script from the MEME suite[54]. It was executed with the following set of arguments: "-order 2 -seed 0 -meme-mod oops -meme-minw 13 -meme-maxw 34 -meme-nmotifs 3 -centrimo-local" and a target database of loxP and loxF8 sequences used at a motif comparison stage. To generate an input file, BEDTools[51] were used to extract 330 bp-long sequences centered at positions of 85 peak summits reported by the ChIP-Seq analysis.

**Recombination assays of ChIP-seq peaks in bacteria and human cells**. Twelve peak sequences were tested in a plasmid-based assay for recombination in bacteria and human cells. A DNA insert (95 bp) around each peak was generated by PCR (primer 23–46) using cycling program 4 and cloned into the pEVO vector via BglII digestion and ligation. After expression of the RecF8 recombinase at 100 μg/ml L-arabinose in E. coli, plasmid extraction was performed and recombination of the peak sequence was analyzed by gel electrophoresis.

In order to test the putative off-target sequences in HEK293T cells, reporter plasmids were generated based on the pCAG-loxPSTOPloxP-ZsGreen (Addgene plasmid # 51269; http://n2t.net/addgene:51269; RRID:Addgene_51269)[55]. The plasmid was modified as described by Lansing et al.[21]. In short, the ZsGreen was exchanged for mCherry and the loxP sites were exchanged with the 95 bp fragment of the 12 different peak hits. Successful recombination will remove a stop cassette, allowing for the expression of mCherry. RecF8 or inactive Cre324 were cloned in the transient mammalian expression vector (EF1a-Rec-P2A-EGFP) and co-transfected to HEK293T cells together with the different peak reporter vectors. Recombination was measured via FACS 48 h after transfection.

**HG2 off-target translocation detection**. Primers were designed around the HG2 off-target (chromosome 15, primers 19 and 20) site and its potential recombination-target site HG2-1 (chromosome 7, primers 21 and 22). PCR products generated from these combinations were sequenced and the presence of the off-targets was validated. Next, a combination of primes as depicted in Supplementary Fig. 16 was used for PCR. In order to validate the sensitivity of the PCR and to have a positive control, a DNA fragment that resembles the translocation product if HG2 is recombined was synthesized. The sensitivity of the PCR was tested by a serial dilution of the DNA fragment ($1:10^2$–$1:10^7$). PCR products of primer combinations $21 + 19$ and $20 + 22$ should reveal a translocation event. PCR program 5 was used for the amplification. The PCR reaction was then carried out on genomic DNA from human cells treated with and without D7 and RecF8. For a positive control genomic DNA of untreated HEK293T cells was mixed with the synthesized translocation fragment for TL7/15 and TL15/7 (Supplementary Fig. 16) in a ratio that would equal 1:200 genome equivalence.

**IPSCs and endothelial cell culturing, differentiation and transfection**. Reprogramming, culturing, and characterization of patient-derived iPSCs (F8 iPSCs) were performed at the Stem Cell Engineering Facility of the Center for Molecular and Cellular Bioengineering (CMCB) at TU Dresden using the CytoTune-iPS 2.0 Sendai Reprogramming Kit (Thermo Fisher Scientific, A16517, Waltham, MA, USA) according to the manufacturer's guidelines. Human iPSCs were cultured in StemFit02 and laminin coating as described above. Endothelial cells were differentiated from F8 iPSCs and maintained in a 6-well dish as described elsewhere[46]. In short, $3 \times 10^5$ iPSCs were seeded for each well on a laminin coated 6-well plate (day 0) using 2 ml of StemFit 02 medium. The first 3 days cells were cultured in the following medium: DMEM F12 with 1%B27, 1% Glutamax, 8 µM CHIR and 25 ng/ml BMP4. The next 3 days cells were cultured in following medium: StemPro34 with 200 ng/ml VEGF and 2 µM Forskolin. Cells were split in on day 7 after seeding. 6-well plates were coated for 1 h at 37 °C with 2 ml PBS containing 10 µg/cm² fibronectin. ECs were washed using 2 ml PBS per well and detached for 2 min at 37 °C using 0.3 ml Trypsin. ECs were collected in DMEM F12 and pelleted for 3 min at 300 × *g*. The pellet was resuspended in EC maintenance medium (StemPro34 with 100 ng/ml VEGF). The cells were split in a ration of 1:4 and seeded on the coated 6-well plates. For each well 700 ng mRNA (250 ng mRNA of each recombinase monomer or 500 ng of the linked RecF8 mRNA and 200 ng mCherry mRNA) and 4 µl Lipofectamine™ MessengerMAX™ was mixed with 100 µl Opti-MEM I Reduced Serum Medium (ThermoFisher, prewarmed at RT) and 4 µl Lipofectamine™ MessengerMAX™. The mRNA and Lipofectamine™ MessengerMAX™ mixtures were combined and incubated 15 min at RT. Afterward the transfection mixture was added to the ECs. Four hours after transfection the EC maintenance medium was renewed and the cells were analyzed 48 h post-transfection.

**Factor VIII qPCR**. RNA was isolated using the RNeasy mini kit (Qiagen). Reverse transcription was carried out using the High-Capacity cDNA Reverse Transcription Kit (Applied Biosystems) according to the manufacturer's protocol. qPCR was carried out using the ABsolute QPCR Mix, SYBR Green, no ROX (Thermo Fisher) on the C1000 Touch CFX96 Real-Time System (BioRad). In order to detect the right boundary of exon 1 and exon 2 after recombinase mediated correction of the genomic inversion, specific primers (47 and 48) in exon 1 and exon 3 were used for quantification. F8 mRNA expression was normalized to TATA-Box Binding Protein (TBP).

**Whole-genome sequencing of ECs**. Genomic DNA was extracted from mRNA transfected patient-specific ECs 72 h post transfection. The average size of extracted DNA was determined using a fragment analyzer (average size 10–50 kb). Libraries were prepared following the manufacturer's instructions with 1.25–1.5 µg using the Ligation Sequencing Kit SQK-LSK109 (Oxford Nanopore Technologies). 1 µl of DNA Control sequence (DCS) was included in the library preparation. Samples were sequenced using the PromethION (Oxford Nanopore Technologies). Runs were set to 72 h, the "non-treated" sample was washed and reloaded after 44 h. "Cre-treated" and "Rec8-treated" flow cells were refueled after 44 h. Sequencing reads were aligned to the reference genome using minimap2 (ver. 2.17)[56] with the following command-line arguments: "–MD -a -x map-ont". The reference sequence was modified from GENCODE Release 26 (GRCh38.p10)[57] to accommodate for the patient-specific chrX inversion between positions 155,007,147 and 155,147,780. The minimap2 output was converted into BAM format with samtools[58] Variant calling was performed with two software packages: Sniffles (ver. 1.0.12)[59] and npInv (ver. 1.28)[60], where the latter was exclusively used for calling inversions. Each sample was processed independently and variants predicted from each sample were merged into a single vcf file using SUR-VIVOR (ver. 1.0.7)[61] with arguments set to "500 1 1 -1 -1 -1". In case of the Sniffles pipeline, the variant calling step was repeated in a forced genotyping mode, using the merged file as a fixed reference, as described on the Sniffles web page (https://github.com/fritzsedlazeck/Sniffles/wiki/SV-calling-for-a-population). Sniffles was executed with "–max_distance 500" parameter, and a "–min_support" parameter set to 3, in case of the first run, or -1 in case of the latter genotyping mode. Inversions were detected by npInv executed with "–min 150 –max 500000 –threshold 1" arguments and merged with SURVIVOR. In case of npInv, the forced genotyping mode is not available. Using bcftools[58], resulting vcf files were converted into a

tab-separated format containing positions of variants and counts of reads supporting either a reference or a variant allele.

Counting and filtering of variants was performed in R using the GenomicRanges package (ver. 1.44.0)[62]. In order to exclude patient-specific variants, all variants detected in recombinase-treated samples having their rearrangement breakpoints overlapping (±500 bp) with breakpoints detected in the non-treated sample were filtered out. To nominate variants potentially caused by recombinase activity, the following additional criteria were applied: a variant has to be either an inversion, a translocation or a deletion; in case of deletions and inversion, a distance between breakpoints has to be at least 150 bp; the total number of reads supporting either a reference or a variant allele has to be more than 10 and a fraction of variant-supporting reads cannot exceed 20%. The WGS data is deposited at the European Genome-phenome Archive (EGA) under the ID EGAS00001005496.

**Factor VIII immunohistochemistry**. iPSCs were cultured in 6-well plates (Corning) and were differentiated into endothelial cells as described earlier. At day 3 cells were fixed in 4% paraformaldehyde (PFA) at room temperature for 20 min, permeabilized and immunostained following standard protocol using anti-Factor VIII antibody (Abcam, ab236284, 1:200) and fluorescently labeled secondary antibody (Invitrogen, Alexa fluor 546 goat anti-rabbit, 1:500). Additionally, nuclei were labeled with Hoechst 33342 (Invitrogen, H3570, 1:5000). Images of cells were automatically captured using an automated Operetta CLS confocal microscope (PerkinElmer) at 20× magnification. Subsequent image capturing was performed using the PerkinElmer Columbus Image Analysis System as previously described[63].

**CRISPR/Cas9 and RecF8 fidelity**. In order to compare the accuracy of recombinase-based inversion with nuclease-based inversion, we tested these two different approaches in patient-specific endothelial cells (ECs). Transfections were performed in a 6-well format as described earlier using mRNAs together with Lipofectamine™ MessengerMAX™. One sample was co-transfected with 500 ng of RecF8 mRNA and 200 ng of mCherry mRNA. The other sample was transfected with 1000 ng Cas9 mRNA (TriLink® BioTechnologies) and 10 pmol of gRNA (5′-GGUCCCCGGGGUUGUGCCCC-3′) targeting the inverted repeat as published by Park et al.[7] Two days after transfection genomic DNA was isolated and analyzed for the inversion accuracy. The recombinase or CRISPR/Cas9 mediated inversion product was amplified using primers $16 + 18$ spanning over the loxF8 target site and the gRNA binding site. The recombinase or CRISPR/Cas9 mediated inversion products were amplified using primers $15 + 16$ or $17 + 18$ spanning over each of the loxF8 target sites and the gRNA binding site. The PCR products were purified using ISOLATE II PCR and Gel Kit (Bioline, Meridian Bioscience), cloned into pMiniT 2.0 plasmid, and transformed into NEB 10-beta *E. coli* using the NEB PCR Cloning Kit (NEB) according to the manufacturer's instructions. For each sample, twenty-two colonies were picked and sent for *E. coli* overnight Sanger sequencing (Microsynth) with the cloning analysis forward primer provided with the kit.

**Statistics and reproducibility**. Statistical analysis was performed using GraphPad Prism 8. Relevant details of the statistical test are provided in the figure legends. Experiments that were further statistically analyzed or quantified were replicated and reproduced independently. Representative gel pictures were not replicated for restriction digests (e.g., activity of recombinase libraries during SLiDE or single clones) or PCR reactions (e.g., PCR-based activity of single clones or orientation-PCR of the loxF8 locus in human cells) if no comparative quantification was performed.

**Reporting summary**. Further information on research design is available in the Nature Research Reporting Summary linked to this article.

## Data availability

The datasets generated during and/or analyzed during the current study are not publicly available due to the current filing process of the patent but are available from the corresponding author F.B. on reasonable request. The patient-specific WGS data are available under restricted access due to data privacy laws, but may be obtained with Data Use Agreements with the Technical University of Dresden, Germany. Researchers interested in access to the data may contact F.B. at frank.buchholz@tu-dresden.de. It can take some months to negotiate data use agreements and gain access to the data. The author will assist with any reasonable replication attempts for two years following publication. After positive request the WGS data access is available and can be accessed at the European Genome-phenome Archive (https://ega-archive.org/) under a Study ID: EGAS00001005496. The ChIP-Seq data generated in this study have been deposited at the NCBI Gene Expression Omnibus (https://www.ncbi.nlm.nih.gov/geo) under a Series ID:GSE159492. Other data generated in this study are provided in the Supplementary information. Source data are provided with this paper.

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

## Acknowledgements

This work was supported, in part, by the European Union (ERC 742133, H2020 UPGRADE 825825—F.B.) the BMBF GO-Bio (031B0633—F.B.), the Else Kröner Fresenius Stiftung and the Bayer Hemophilia Awards Program (BHAP—F.B.). Furthermore, we thank the DRESDEN-concept Genome Center at the CMCB from the TU Dresden for performing the sequencing of the ChIP samples. We thank the Flow Cytometry Core Facility and Stem Cell Engineering Core Facility of the CMCB Technology Platform at TU Dresden for their excellent support. Moreover, we thank the Long Read Team of the DRESDEN-concept Genome Center, DFG NGS Competence Center, part of the Center for Molecular and Cellular Bioengineering (CMCB), TU Dresden and MPI-CBG. We also want to thank the Department of Pediatric Hematology and Oncology at University Hospital Dresden for providing samples from hemophilic patients. Lastly, we want to acknowledge and thank the hemophilic donor and his family for providing blood samples to generate the induced pluripotent stem cells.

## Author contributions

F.L., J.S., P.M.S., and J.K. performed the directed evolution of the new recombinases. L.M., F.L., and T.R-R. designed, planned performed the linker development. T.R-R. developed and performed the recombinase ChIP-Seq. experiment. F.L., M.K., and K.I. performed the iPSCs and the EC differentiation work in the US. F.L., T.G., and N.R-M. worked with the patient-derived iPSCs in Germany. J.H. performed structure-based analyses and evaluated data. M.P-R. and L.T.S. contributed to the bioinformatic analysis and the target site identification. M.P-R. processed the ChIP-Seq. data and WGS data. R.K. organized and transferred the patient-specific cells. C.G. reprogrammed patient-specific cells to iPSCs. F.B. and T.T. designed the study, directed the study, evaluated data and wrote the manuscript. F.L. analyzed the data and wrote the manuscript.

## Competing interests

Technical University (Technische Univeristät) Dresden has filed a patent application (WO/2021/110846) based on this work, in which F.L., L.M., T.R-R., J.S., J.K., and F.B. are listed as inventors. The remaining authors declare no competing interests.
