## [Peer Review File · Nature Communications]

Reviewers' Comments:

Reviewer #1:

Remarks to the Author:

After reading all previous reviewers' comments and authors' responses, I believe this paper addressed all major concerns and much improved. I am for its publication.

Reviewer #2:

Remarks to the Author:

I have carefully reviewed the authors' response and am pleased with the changes that have been made in the revised manuscript.

Reviewer #3:

Remarks to the Author:

This manuscript by Lansing et al describes a directed evolution approach to generate a novel recombinase to specifically correct a 140 Kb-long genomic inversion in F8 causing Hemophilia A. This study is a follow up of the authors former SLiDE technology. Here the technology is exploited to generate a designer recombinase, RecF8, which can accurately correct the F8 genetic defect in Hemophilia A patient derived cells. The RecF8 tool is further optimized in this work through linker evolution to minimize off-target effects. Proof of concepts on the efficacy of the strategy is obtained through phenotypic and genotypic assays including patient-derived iPSC cells underscoring the therapeutic potential of RecF8.

This is an elegant approach further enriching the toolbox for genome editing. Yet since similar studies were published using talens (PNAS 2014) or CRISPR-Cas (Cell stem cell 2015) it is recommendable addressing the advantages of by this tool.

Major comments

1. Comparison of the technology here reported with CRISPR approach by Park et al (cell stem cells 2015) is limited to the sequence analysis in 22 clones, highlighting the percentages of indels produced by CRISPR technology (fig. 6). Yet the study does not report the efficiency of CRISPR editing versus RecF8 (frequencies of editing using the two technologies). This piece of information is relevant to of the scientific community to have a full picture of the two technologies in this context (and potentially extendable to other targets).

Reviewer #1 (Remarks to the Author):

After reading all previous reviewers' comments and authors' responses, I believe this paper addressed all major concerns and much improved. I am for its publication.

Response: Thank you very much for the positive feedback. We were very happy for the constructive feedback during the revision process.

Reviewer #2 (Remarks to the Author):

I have carefully reviewed the authors' response and am pleased with the changes that have been made in the revised manuscript.

Response: Thank you very much for your valuable input during the revision process.

Reviewer #3 (Remarks to the Author):

This manuscript by Lansing et al describes a directed evolution approach to generate a novel recombinase to specifically correct a 140 Kb-long genomic inversion in F8 causing Hemophilia A. This study is a follow up of the authors former SLiDE technology. Here the technology is exploited to generate a designer recombinase, RecF8, which can accurately correct the F8 genetic defect in Hemophilia A patient derived cells. The RecF8 tool is further optimized in this work through linker evolution to minimize off-target effects. Proof of concepts on the efficacy of the strategy is obtained through phenotypic and genotypic assays including patient-derived iPSC cells underscoring the therapeutic potential of RecF8. This is an elegant approach further enriching the toolbox for genome editing. Yet since similar studies were published using talens (PNAS 2014) or CRISPR-Cas (Cell stem cell 2015) it is recommendable addressing the advantages of by this tool.

Major comments

1. Comparison of the technology here reported with CRISPR approach by Park et al (cell stem cells 2015) is limited to the sequence analysis in 22 clones, highlighting the percentages of indels produced by CRISPR technology (fig. 6). Yet the study does not report the efficiency of CRISPR editing versus RecF8 (frequencies of editing using the two technologies). This piece of information is relevant to of the scientific community to have a full picture of the two technologies in this context (and potentially extendable to other targets).

Response: We thank reviewer 3 for the comment. Indeed, we agree that the comparison of the inversion efficiency from the two different approaches is important. Therefore, we have already included a direct citation of the results on the inversion efficiency obtained by Park et al. in the last revised manuscript version. 'It has been shown that F8 gene inversions can be corrected using nucleases such as TALENs or CRISPR/Cas9^{6,7}. Reported efficiencies of up to 6,7% inversion rates are in line with our RecF8-mediated results.'

Moreover, the CRISPR mediated inversion efficiencies in HeLa cells and iPSCs are already thoroughly examined in the study by Park et al.

1. *'In addition, RGEN 01 induced the inversion of the 140-kbp chromosomal segment in HeLa cells, as shown by inversion-specific PCR (Figure S1C). The frequency of this inversion ranged from 2.2% to 3.1%, as measured by digital PCR (Table S1)'*
2. *'Eight colonies (not necessarily derived from single cells) out of 120 colonies (6.7%) produced positive PCR bands on an agarose gel. Four colonies were then further cultured to obtain single-cell-derived clones. These clones produced PCR amplicons corresponding to the int1h-1 and int1h-2 regions, indicating that the inverted 140-kbp chromosomal segment in Pa1 cells was reverted (Figure S1E).'*

Therefore, we focused in our manuscript on the fidelity of the two different approaches and reported the indel frequencies at the edited locus using either CRISPR/Cas9 or RecF8. We think, that the information we provide by the citations (Park et al. 2014 and Park et al. 2015) together with our data on the fidelity, is informative for the scientific community to have a full picture of the two technologies in this context.